# Environmental Footprint Assessment of a Cleanup at Hypothetical Contaminated Site

**Muhammad Azhar Ali Khan** [1], **Zakria Qadir** [2,*], **Muhammad Asad** [1], **Abbas Z. Kouzani** [3]
**and M. A. Parvez Mahmud** [3]

[1] Mechanical Engineering Department, College of Engineering, Prince Mohammad bin Fahd University, Al-Khobar 31952, Saudi Arabia; mkhan6@pmu.edu.sa (M.A.A.K.); masad@pmu.edu.sa (M.A.)
[2] School of Computing Engineering and Mathematics, Western Sydney University, Locked Bag 1797, Penrith 2751, Australia
[3] School of Engineering, Deakin University, Geelong 3216, Australia; abbas.kouzani@deakin.edu.au (A.Z.K.); m.a.mahmud@deakin.edu.au (M.A.P.M.)
\* Correspondence: z.qadir@westernsydney.edu.au

**Abstract:** Contaminated site management is currently a critical problem area all over the world, which opens a wide discussion in the areas of policy, research and practice at national and international levels. Conventional site management and remediation techniques are often aimed at reducing the contaminant levels to an acceptable level in a short period of time at low cost. Owing to the fact that the conventional approach may not be sustainable as it overlooks many ancillary environmental effects, there is an immense need of "sustainable" or "green" approaches. Green approaches address environmental, social and economic impacts throughout the remediation process and are capable of conserving the natural resources and protecting air, water and soil quality through reduced emissions and other waste burdens. This paper presents a methodology to quantify the environmental footprint of a cleanup for a hypothetical contaminated site by using the US Environmental Protection Agency's (EPA) Spreadsheet for Environmental Footprint Assessment (SEFA). The hypothetical contaminated site is selected from a metropolitan city of Pakistan and the environmental footprint of the cleanup is analyzed under three different scenarios: cleanup without any renewable energy sources at all, cleanup with a small share of renewable energy sources, and cleanup with a large share of renewable energy sources. It is concluded that integration of renewable energy sources into the remedial system design is a promising idea which can reduce $CO_2$, $NO_x$, $SO_x$, PM and HAP emissions up to 68%.

**Keywords:** environmental footprint; cleanup; green remediation; renewable energy sources

## 1. Introduction

Over the last few years, a rapid increase is observed in the awareness and dialogue about the environment in general and, in particular, about the issues, such as sustainability, recycling, greenhouse gas (GHGs) emissions, and a greener world [1–4]. In almost all spheres of life, people take a keen interest in understanding how goods are produced, how they are delivered and, eventually, how it all impacts the environment [5]. Within this context, the restoration of contaminated and toxic places is now also identified as a critical problem, which opens a wide discussion in the areas of policy, research, and practice at national and international levels [6–11]. At this point, the following question arises: "Is there any environmentally friendly way to clean the environment?" The US Environmental Protection Agency (EPA) tries to answer this question by adding a new phrase to today's environmental lexicon: Green Remediation [12,13].

Conventional and Green remediation are in contrast to each other in many perspectives. A conventional site remediation approach is normally based on (a) the effectiveness and appropriateness of the particular remediation method to meet the remedial goals; (b) ease of implementation; (c) remediation costs; and (d) remediation timeframe [14].

Green remediation, on the other hand, employs the idea of protecting human health and environment while minimizing the environmental side effects. It asks for (a) efficient use of natural resources and energy; (b) reduction in the negative impacts on the environment; (c) minimization or elimination of pollution at its source; and (d) reduction in waste to greatest possible extent [14].

Owing to the fact that the conventional approaches may not be sustainable as they overlook many ancillary environmental effects, there is an immense need of "Sustainable" or "Green" approaches which address environmental, social and economic aspects throughout the remediation process. One way of making the remediation process green could be analyzing the extent to which it is impacting the environment. Such analyses in a green remediation setting are often termed Environmental Footprint Analysis. Primarily, an environmental footprint highlights the aspects of a cleanup that dominates the footprint and then provides the opportunity to improve the remedy efficiency and effectiveness by implying a range of alternatives to the existing remedial system.

Incorporating renewable energy sources into a cleanup activity can offer increased sustainability and long-term cost savings [15]. The use of renewable energy in remedial system designs is not a new idea, and it is already observed at various sites in the world. Amanda [15] identifies solar, wind, landfill gas, and biodiesels as the possible options that can be integrated into a remedy. The Saint St. Croix Alumina site in the Virgin Islands uses wind-driven turbine compressors (WDTC) to drive hydraulic oil "skimmer" pumps to recover free product (oil) from ground water. The system does not produce electricity to power pumps; instead, it uses compressed air generated by WDTCs. PV arrays and wind-driven electricity generators are also employed to power submersible pumps for oil, groundwater and petroleum hydrocarbon recovery [15]. Another application of renewable energy in remedial systems is the use of solar energy to pump water into and circulate through a bioreactor installed at the Altus Air Force Base in Oklahoma, to remove TCE from ground water. This project is found to be cost effective in terms of avoiding construction of a power transmission line from the utility grid to the bioreactor's remote location [16]. The BP Paulsboro, a former petroleum and specialty-chemical storage facility, is being remediated by a large onsite pump and treat (P&T) system which is empowered by solar energy since 2003. Almost 20–25% electricity requirements are met by solar energy and for the rest the system relies on electricity from the utility grid. A reduction in the emissions of $CO_2$ by 571,000 pounds per year, sulfur dioxide ($SO_2$) by 1600 pounds per year, and nitrogen oxide (NOx) by 1100 pounds per year is expected from this hybrid remedial system design [17].

The EPA's Spreadsheet for Environmental Footprint Assessment (SEFA) is among the key tools available for such investigations. However, very limited published literature is available about using SEFA for environmental impact assessment of remediation of contaminated sites. Marco et al. [18] study the environmental impact of the remediation of an aquifer below an industrial site in the Bologna area. Three proposed systems were investigated for environmental impact using two environmental footprint analysis tools, i.e., SiteWise$^{TM}$ and SEFA. The three solutions studied and compared for environmental footprint are (i) groundwater extraction system, treatment and reinjection, (ii) reductive bioremediation, and (iii) in situ chemical oxidation (ISCO). Based on the results obtained from both tools, bioremediation is found to be the appropriate remediation with minimum GHG and lower levels of environmental impacts. In addition, the higher environmental impacts caused by ISCO is due to frequent multiple injection events with Potassium Permanganate, in contrast with the general single injection performed with bioremediation.

In this paper, the environmental footprint of a cleanup at a hypothetical contaminated site is studied by using EPA's Spreadsheet for Environmental Footprint Assessment (SEFA). The underlying idea is to develop a methodology for analyzing the environmental impact of a cleanup of any contaminated site using this easily accessible tool. Therefore, a hypothetical contaminated site is selected from a metropolitan city in Pakistan and the environmental footprint of the cleanup is analyzed under three different scenarios: cleanup

without any renewable energy sources at all, cleanup with a small share of renewable energy sources, and cleanup with a large share of renewable energy sources.

## 2. Core Elements of Green Remediation

Green remediation aims to minimize the energy and environmental footprint of a site remediation and revitalization. A set of core elements is made by the US EPA [19] as shown in Figure 1, which actually describes the potential areas that can reduce the environmental footprint of a site cleanup. The details of these core elements are provided in the following sub-sections.

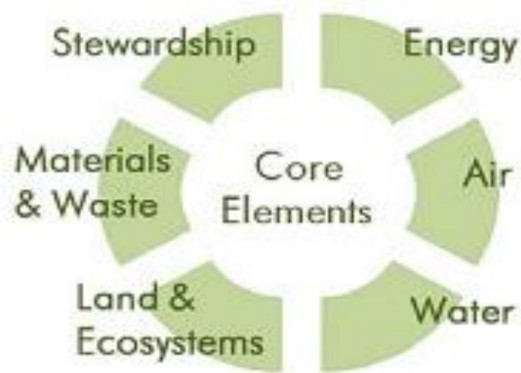

**Figure 1.** Core elements of green remediation [18].

### 2.1. Energy

The energy requirement of the treatment system is extremely important to analyze in terms of green remediation. This element emphasizes the use of passive energy sources in order to meet all remediation objectives. In addition to that, energy efficient equipment should be used and maintained at peak performance to maximize efficiency. Moreover, periodical evaluation and optimization of energy efficiency of equipment with high energy demand can significantly reduce the energy consumption. Besides all this, the integration of renewable energy systems can replace or at least offset the electricity requirements otherwise met by the utility grid.

### 2.2. Air

This element is mostly concerned with the air emissions caused by different types of fuel in any onsite or offsite operation of a cleanup. It emphasizes to minimize the use of heavy equipment requiring high amounts of fuels and to use cleaner fuels for the operation of these equipment. It also takes into account the reduction of toxic and priority pollutants, such as ozone, particulate matter, carbon monoxide, nitrogen dioxide, sulfur dioxide, and lead, with the minimization of dust export of the contaminants.

### 2.3. Water

Water requirement and the impacts on water resources is also a key component of green remediation by minimizing the freshwater use and maximize the water reuse during daily operations and treatments processes. The treated water can be reclaimed for beneficial use such as irrigation. Moreover, the nearby water bodies should be prevented from impacts such as nutrient loading.

### 2.4. Land and Ecosystems

In terms of land and ecosystems impacts, the minimum invasive in situ technologies should be used and passive energy technologies like bioremediation and phytoremediation should be selected as primary remedies where possible and effective. This component

also calls for the minimization of soil and habitat disturbance and reduction in noise and lighting disturbance.

### 2.5. Materials and Wastes

For green remediation, the selected technologies should be capable of generating minimum waste. Re-use and recycling of material generated at or removed from the site should be promoted. A major concern in this component is the minimization of natural resource extraction and disposal. If feasible, passive sampling devices should be used that produce minimum waste.

### 2.6. Stewardship

The stewardship goals are usually long-term such as reduction of $CO_2$, $N_2O$, $CH_4$, and other greenhouse gases emissions contributing to climate change, integration of an adaptive management approach into controls for a site, installation of renewable energy systems for cleanup and future activities on redeveloped land, and community involvement to increase public acceptance and awareness of long-term activities and restrictions.

## 3. Environmental Footprint of a Cleanup Project

The term "footprint" refers to the quantification of a specific parameter that has been assigned a particular meaning. For example, in terms of "carbon footprint", it is the quantification of carbon dioxide (and other GHGs) emitted into the air by a particular activity, facility, or individual. Green remediation should be analyzed in detail to closely examine the components of the remedial system and to identify the large contributors to the environmental footprint. The purpose, limitations, value, and the level of effort and cost for an environmental footprint assessment are discussed in the following subsections.

### 3.1. Purpose

The first and foremost purpose of environmental footprint analysis is to facilitate the implementation of EPA's principles for greener cleanups [8]. By doing this analysis, the quantification of metrics for a cleanup can be done and a set of technical suggestions can be made on the approaches to reduce the footprint of a remedial system.

### 3.2. Limitations

The environmental footprint assessment is not intended to be a detailed life-cycle analysis (LCA). It uses a suitable number of green remediation metrics to represent the core elements of green remediation but limits the number of metrics to streamline the footprint analysis process. It is also limited in a sense that it is not a mandatory requirement of EPA but it is just intended to support the remedial process and to reduce the environmental impact of a cleanup activity.

### 3.3. Value

The environmental footprint assessment can be considered as a valuable component in a cleanup because it can quantify the footprint reductions of a cleanup project. The dominant aspects of the footprint can be highlighted and thus strategies can be adapted to reduce their contributions in the footprint. Based on its results, it also provides the opportunity to improve the remedy efficiency and effectiveness which is usually a missing element in a more conventional evaluation.

### 3.4. Level of Effort and Cost

According to the US EPA, the environmental footprint analysis adds negligible amount to the level of effort or cost for an overall remediation and a fraction of any particular remedial activity, such as a remedy design or an optimization evaluation. EPA expects to have an addition of 10% to the level of effort or cost of an optimization evaluation, or less than 5 percent to the level of effort or cost of a remedial design [20]. Footprint evaluation

mainly focuses on green remediation metrics and does not quantify the cleanup costs. Since the cleanup costs are directly related to core elements of greener cleanups, the cost savings can be expected over the life of a cleanup project. However, these reductions are project-specific and are highly related to the location and time span of a remedial operation.

## 4. Method and Material

### 4.1. Site Selection

The Sustainable Development Policy Institute (SDPI) in collaboration with the Blacksmith Institute (BSI), USA, carried out a Global Inventory Project (GIP) for the mapping of chemical contaminated sites in Pakistan, to improve public health and environment in and around the public site area. SDPI identified a total of 31 contaminated sites in Pakistan [21], the distribution of which is shown in Figure 2.

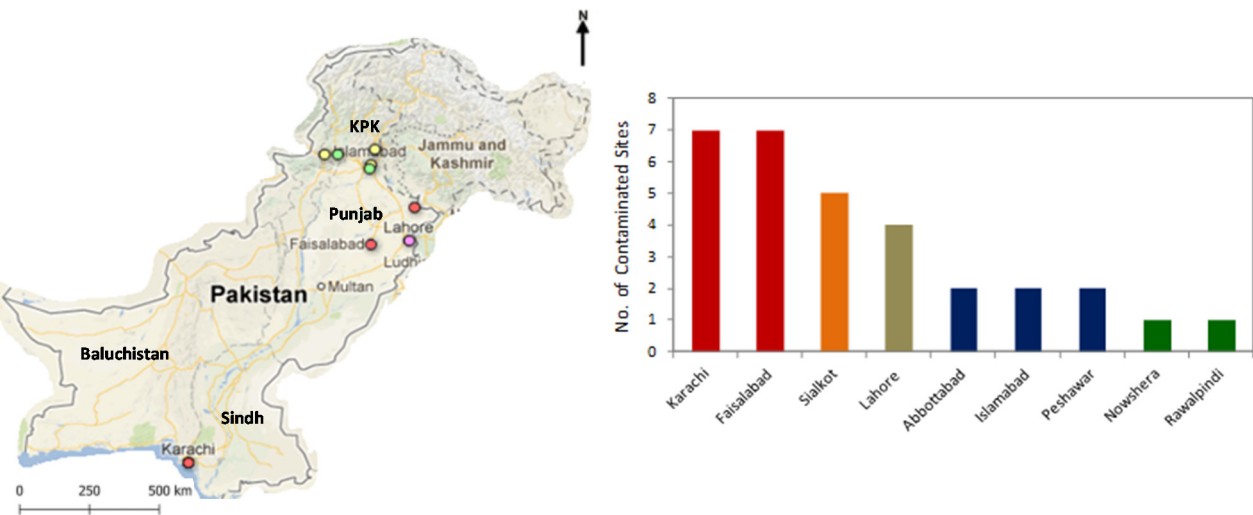

**Figure 2.** Distribution of contaminated sites in Pakistan.

In general, the two provinces Khyber Pakhtunkhwa (KPK) and Punjab are found to have maximum concentration of contaminated sites, as shown in the map of Pakistan. Since Karachi and Faisalabad are considered to be the industrial hubs of Pakistan, each of them contains seven contaminated sites. For the purpose of this study, a contaminated site is selected from Karachi. The site is potentially contaminated due to a number of industries in the vicinity. Moreover, the site is near to the creek. Thus, it is expected to have both groundwater and surface water pollution.

### 4.2. Hypothetical Situation

A pump and treat system are under design to treat LNAPL (petroleum product) contamination caused by a spill from an underground tank. An oil/water separation technique will be used for cleanup. The clean water is then discharged to the creek. A schematic of the LNAPL release and subsequent migration and the remedial system is presented in Figure 3a,b, respectively.

The construction of the remedial system will include:

i.      Ten 6-inch extraction wells, each to 60 feet deep with 20-foot screens;
ii.     3000 feet of 6-inch HDPE piping with electrical conduit and wiring;
iii.    80 ft × 100 ft building that is 30 feet high;
iv.     200 ft × 200 ft reinforced concrete pad and containment area (20,000 ft$^3$ of concrete).

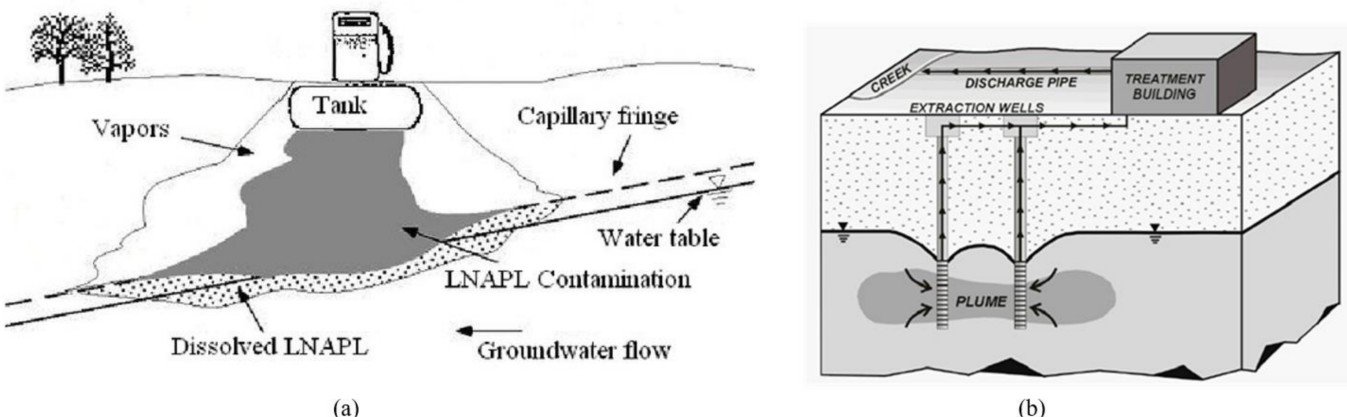

**Figure 3.** (**a**) LNAPL release and subsequent migration [22] and (**b**) pump and treat remedial system.

The data required to analyze the environmental footprint of this remedial system will include the details of materials used, wastes generated, water consumption, energy consumption and air emissions. The largest contributor to refined materials is expected to be the building steel (292,000 lbs) over a 30-year period. The largest contributor for unrefined materials is expected to be the aggregate in the concrete for the building foundation (about 1200 tons). No specific appreciable non-hazardous waste streams have been identified. The dewatered sludge is expected to be 2600 tons. The project team has chosen a percent-based screening limit of 1 percent for refined and unrefined materials and magnitude-based limits of 1000 lbs for refined materials and 1 ton for unrefined materials and wastes. Table 1 lists all the data which are provided to the EPA's Spreadsheet for Environmental Footprint Assessment.

*4.3. Scenarios for Analysis*

The environmental footprint of the cleanup at the hypothetical site is discussed in Sections 4.1 and 4.2 and is analyzed under three different scenarios.

4.3.1. Cleanup without Any Renewable Energy Resources at All

In this scenario, it is assumed that all of the energy used in the remedial systems, which is estimated to be 33,000,000 kWh, is taken from the grid for which the fuel mix is already described in Table 1. Neither the renewable energy is generated onsite nor it is voluntarily purchased from a renewable energy producer. It is important to note that the fuel mix for grid electricity in Pakistan is dominated by conventional fossil fuels such as oil and natural gas.

4.3.2. Cleanup with a Small Share of Renewable Energy Sources

For this case, it is assumed that the energy demands in remedial operations are met by employing a small share of renewable energy. This means that 70% of the electricity is taken from grid while 30% from renewable energy resources. Out of this 30%, 20% electricity is generated onsite using renewable energy sources and 10% is voluntarily purchased from a renewable energy producer.

4.3.3. Cleanup with a Large Share of Renewable Energy Sources

This scenario assumes to have a large share of renewable energy resources in order to meet the energy demands of remedial operations. In this case, only 30% electricity is taken from grid while 70% from renewable energy resources. Out of this 70%, 50% electricity is generated onsite using renewable energy sources and 20% is voluntarily purchased from a renewable energy producer.

**Table 1.** Manual data input to EPA's spreadsheet for environmental footprint assessment of a cleanup.

| Material and Use | Units | Quantity | Conversion Factor to lbs | % Recycled or Reused Content | Quantity (lbs) | |
|---|---|---|---|---|---|---|
| | | | | | Virgin | Recycled |
| Refined Materials | | | | | | |
| Well-PV casing and grout | | | | | 0 | 0 |
| Wells-Screen | | | | | 0 | 0 |
| Piping and Conduit | ft | 3000 | 7.5 | 0% | 22,500 | 0 |
| Building Steel | ft$^3$ | 240,000 | 1 | 55% | 108,000 | 132,000 |
| Concrete Reinforcing Steel | ft$^2$ | 40,000 | 1.3 | 55% | 23,400 | 28,600 |
| Cement Portion of Concrete | ft$^3$ | 20,000 | 22 | 20% | 352,000 | 88,000 |
| Process Equipments | | | | | 0 | 0 |
| Process Controls | | | | | 0 | 0 |
| | | | | | 0 | 0 |
| Unrefined Materials | | | | | | |
| Well-Sand Pack | | | | | 0 | 0 |
| Aggregate for Concrete | ft$^3$ | 20,000 | 0.0575 | 0% | 1150 | 0 |
| Waste Disposal (tons) | | | | | | |
| **Hazardous Waste** | | | | | | |
| 2600 tons of Hazardous Waste in the form of Sludge | | | | | | 2600 |
| Water Usage | | | | | | |
| Water Resource | | Description of Quality of Water Used | | Volume Used (1000 gallons) | Uses | Fate of Used Water |
| **Extracted groundwater #1** | | | | | | |
| Location: | | Shallow Aquifer, Marginal Quality | | 11,000,000 | Treatment | Creek |
| Aquifer: | | | | | | |
| Labor, Mobilizations, Mileage, and Fuel | | | | | | |
| Participant | | Crew Size | Number of Days Worked | Hours Worked Per Day | Total Hours Worked | Number of Roundtrips to Site |
| SGS Pakistan | | 20 | 90 | 8 | 14,400 | 100 |
| **Roundtrip Miles to Site** | **Mode of Transport.** | | **Fuel Type** | **Total Miles** | **Fuel Usage Rate** | **Total Fuel Used (gal)** |
| 7.2 | Bus | | Diesel | 720 | 96 | 8 |

**Table 1.** *Cont.*

| Material and Use | Units | Quantity | Conversion Factor to lbs | % Recycled or Reused Content | Quantity (lbs) | |
|---|---|---|---|---|---|---|
| | | | | | Virgin | Recycled |
| On-Site Equipment Use, Mobilization, and Fuel Usage | | | | | | |
| Equipment Type * | | HP | Load Factor | Equip. Fuel Type | Units of Fuel Used per Hour | Total Hours Operated |
| Drilling-medium rig (150 HP) | | 150 | 1% | Diesel | 0.05625 | 320 |
| Gallons of Fuel Used On-Site | Number of Roundtrips to Site | Roundtrip Miles to Site | Total Miles Transported | Transport Fuel Type | Fuel Usage Rate | Total Fuel Used for Transport (gal) |
| 18 | 1 | 7.2 | 7.2 | Diesel | 6 | 1.2 |
| On-Site Electricity Use | | | | | | |
| Equipment Type | HP | % Full Load | Efficiency (%) | Electrical Rating (kW) | Hours Used | Energy Used (kWh) |
| Six 0.75 hp extraction pump | 4.5 | 80% | 65% | 4.131692308 | 1800 | 7437.046154 |
| Two 1 hp discharge pumps | 2 | 80% | 75% | 1.591466667 | 1800 | 2864.64 |
| On-Site Natural Gas Use | | | | | | |
| Equipment Type | | Power Rating (btu/hr) | Efficiency | Total Hours Used | Btus of Gas Required | Total ccf Used |
| Building Heat | | 200,000 | 80% | 2000 | 500,000,000 | 4854.368932 |
| Materials Use (including Potable Water) and Transportation | | | | | | |
| Material Type or Public Water | | | Unit | Quantity | Tons | Default One-Way Miles |
| Cement | | | dry-lb | 440,000 | 220 | 500 |
| Steel | | | lb | 292,000 | 146 | 500 |
| HDPE | | | lb | 22,500 | 11.25 | 500 |
| Concrete | | | lb | 2,300,000 | 1150 | 25 |
| Site-Spec. One-Way Distance (miles) * | Number of One-way Trips to Site | Mode of Transport. | | Fuel Type | Fuel Usage Rate (gptm or mpg) | Total Fuel Used (gallons) |
| 25 | 1 | Truck (mpg) | | Diesel | 6 | 4.2 |
| 25 | 1 | Truck (mpg) | | Diesel | 6 | 4.2 |
| 25 | 1 | Truck (mpg) | | Diesel | 6 | 4.2 |
| 25 | 1 | Truck (mpg) | | Diesel | 6 | 4.2 |

**Table 1.** *Cont.*

| Material and Use | Units | Quantity | Conversion Factor to lbs | % Recycled or Reused Content | Quantity (lbs) | |
|---|---|---|---|---|---|---|
| | | | | | Virgin | Recycled |
| **Waste Transportation and Disposal** | | | | | | |
| | Waste Destination | | Unit | Quantity | Tons | Default One-Way Miles |
| | Hazardous waste landfill | | tons | 2600 | 2600 | 500 |
| Site-Spec. One-Way Distance (miles) | Number of One-way Trips to Site | Mode of Transport. | | Fuel Type | Fuel Use Rate (gptm or mpg) | Total Fuel Use (gallons) |
| 30 | 1 | Truck (mpg) | | Diesel | 6 | 5 |
| **Fuel Mix for Grid Electricity** | | | | | | |
| | Type | | | % of Total Used | | |
| *Conventional Energy* | | | | | | |
| Coal | | | | 0% | | |
| Natural Gas | | | | 27% | | |
| Oil | | | | 34% | | |
| Nuclear | | | | 6% | | |
| Biomass | | | | 0% | | |
| Geothermal | | | | 0% | | |
| Hydro | | | | 33% | | |
| Solar | | | | 0% | | |
| Wind | | | | 0% | | |
| Other (enter information below) | | | | 0% | | |
| **Total** | | | | **100%** | | |

### 4.4. Green Remediation Metrics

The green remediation metrics are mainly based on five out of six core elements. These metrics provide an opportunity to the remedial team to change their strategies for environmental benefit. The details of these green remediation metrics are presented in Table 2.

**Table 2.** Green remediation metrics (modified from [20]).

| Core Element | | Metric | Unit of Measure |
|---|---|---|---|
| Materials and Waste | M&W-1 | Refined materials used on site | tons |
| | M&W-2 | Percent of refined materials from recycled or waste material | percent |
| | M&W-3 | Unrefined materials used on site | tons |
| | M&W-4 | percent of unrefined materials from recycled or waste material | pecent |
| | M&W-5 | Onsite hazardous waste generated | tons |
| | M&W-6 | Onsite non-hazardous waste generated | tons |
| | M&W-7 | Percent of total potential onsite waste that is recycled or reused | percent |
| Water | | Onsite water use (by source) | |
| | W-1 | Source: Groundwater, Purpose: Treatment, Fate: Creek | millions of gallons |
| Energy | E-1 | Total energy use | MMBtu |
| | E-2 | Total energy voluntarily derived from renewable resources | |
| | E-2A | - Onsite generation or use and biodiesel use | MMBtu |
| | E-2B | - Voluntary purchase of renewable electricity | MWh |
| | E-2C | - Voluntary purchase of RECs | MWh |
| Air | A-1 | Onsite Nox, Sox, and PM emissions | lbs |
| | A-2 | Onsite HAP emissions | lbs |
| | A-3 | Total Nox, Sox, and PM emissions | lbs |
| | A-4 | Total HAP emissions | lbs |
| | A-5 | Total GHG emissions | tons $CO_2$-e |
| Land and Ecosystem | | Qualitative Description | |

#### 4.4.1. Materials and Waste Metrics

The material metrics takes into account the total amount of materials used onsite and the percentage of those materials that are produced from recycled material, reused material, or waste material. The waste metrics consider the total amount of waste generated on site and the percentage of total potential onsite waste that is recycled or reused [19].

#### 4.4.2. Water Metrics

According to [8], the water metrics include the source and amount of water used onsite, and the fate of water after use. The sources of water could be public potable water supply, ground water from a local aquifer, surface water, and reclaimed water, etc. In a remedial system, water is mainly used in equipment decontamination, treatment, injection for plume migration, and chemical blending, etc. In terms of fate of used water, it can be discharged to groundwater and fresh surface water, used in irrigation and industrial processes, or reused in a public or domestic water supply.

#### 4.4.3. Energy Metrics

The energy metrics are mainly focused on the total amount of energy used in the remedial operation (both onsite and offsite). Moreover, they also take into account the total amount of renewable energy used in a remedial operation, such as onsite generation and

use of renewable energy and use of biodiesel, voluntary purchase of renewable electricity and renewable energy certificates.

### 4.4.4. Air Metrics

The air metrics consider emissions of GHGs, nitrogen oxides (NOx), sulfur oxides (SOx), particulate matter less than 10 microns in size (PM10), and hazardous air pollutants (HAPs) [21].

### 4.4.5. Land and Ecosystem

This metric is about a qualitative description of potential disturbance in land and ecosystem which will be caused by the employed remediation technique.

### *4.5. Footprint Methodology*

The footprint methodology is actually based on seven-steps as shown in Figure 4. The evaluation of footprint begins with setting goals and scope of the analysis followed by gathering the information of the remedial system to be footprinted. Based on this information, the quantification of onsite materials metrics, waste metrics and water metrics is carried out. The materials, waste and water information, and other remedy information, then helps quantify the energy and air metrics. A qualitative description of the ecosystems, which will be disturbed by the implementation of remedial system, is also included in the analysis. Finally, the results are presented and analyzed for the identification of large contributors to the metrics and for the opportunities to reduce overall footprint of the remedial system.

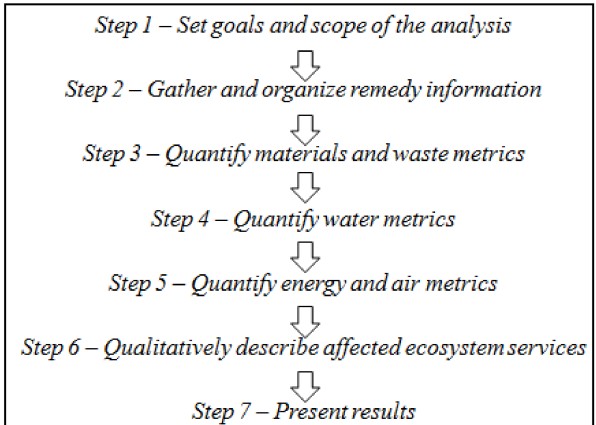

**Figure 4.** Steps in footprint methodology (modified from [8]).

### 5. Results and Discussions

In Table 3, the summary of environmental footprint of the cleanup under all three scenarios is presented. Figures 5–7 represent the contributions of different elements of the cleanup in environmental footprint in terms of total energy, $CO_2$ emissions, and NOx, SOx, PM, and HAP emissions, respectively. Table 4 lists the reductions in $CO_2$, HAP, and NOx, SOx, and PM emissions as a result of renewable energy integration into the remedial system as discussed above in Section 4.3.

**Table 3.** Summary of the environmental footprint of cleanup under all scenarios.

| Core Element | | Metric | Unit of Measure | Scenario 1 | Scenario 2 | Scenario 3 |
|---|---|---|---|---|---|---|
| **Materials and Waste** | M&W-1 | Refined materials used on site | tons | 377 | 377 | 377 |
| | M&W-2 | Percent of refined materials from recycled or waste material | percent | 33 | 33 | 33 |
| | M&W-3 | Unrefined materials used on site | tons | 1150 | 1150 | 1150 |
| | M&W-4 | percent of unrefined materials from recycled or waste material | pecent | 0 | 0 | 0 |
| | M&W-5 | Onsite hazardous waste generated | tons | 2600 | 2600 | 2600 |
| | M&W-6 | Onsite non-hazardous waste generated | tons | 0 | 0 | 0 |
| | M&W-7 | Percent of total potential onsite waste that is recycled or reused | percent | 0 | 0 | 0 |
| **Water** | | Onsite water use (by source) | | | | |
| | W-1 | Source: Groundwater, Purpose: Treatment, Fate: Creek | millions of gallons | 110,000 | 110,000 | 110,000 |
| **Energy** | E-1 | Total energy use | MMBtu | 408,551 | 309,966 | 182,275 |
| | E-2 | Total energy voluntarily derived from renewable resources | | | | |
| | E-2A | - Onsite generation or use and biodiesel use | MMBtu | 0 | 22,526 | 56,315 |
| | E-2B | - Voluntary purchase of renewable electricity | MWh | 0 | 3300 | 6600 |
| | E-2C | - Voluntary purchase of RECs | MWh | 0 | 0 | 0 |
| **Air** | A-1 | Onsite Nox, Sox, and PM emissions | lbs | 55 | 55 | 55 |
| | A-2 | Onsite HAP emissions | lbs | 0 | 0 | 0 |
| | A-3 | Total NOx, SOx, and PM emissions | lbs | 219,726 | 155,511 | 69,891 |
| | A-4 | Total HAP emissions | lbs | 1119 | 801 | 377 |
| | A-5 | Total GHG emissions | tons $CO_2$-e | 41,741,757 | 29,609,307 | 13,432,707 |
| **Land and Ecosystems** | | Land and Ecosystem will be disturbed in terms of hazardous waste disposal offsite, depending upon the technique to be used in its disposal. | | | | |

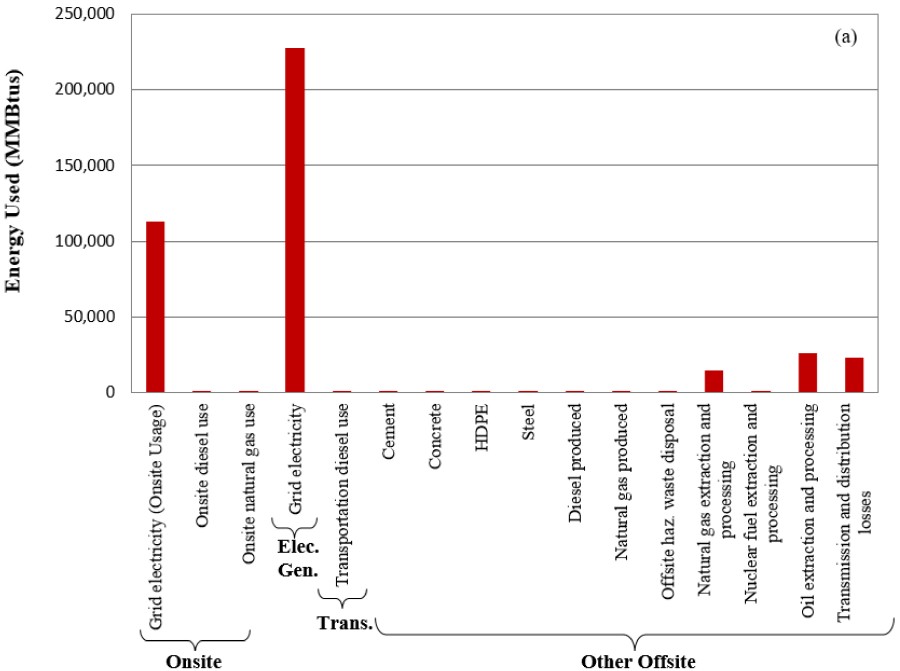

**Figure 5.** *Cont.*



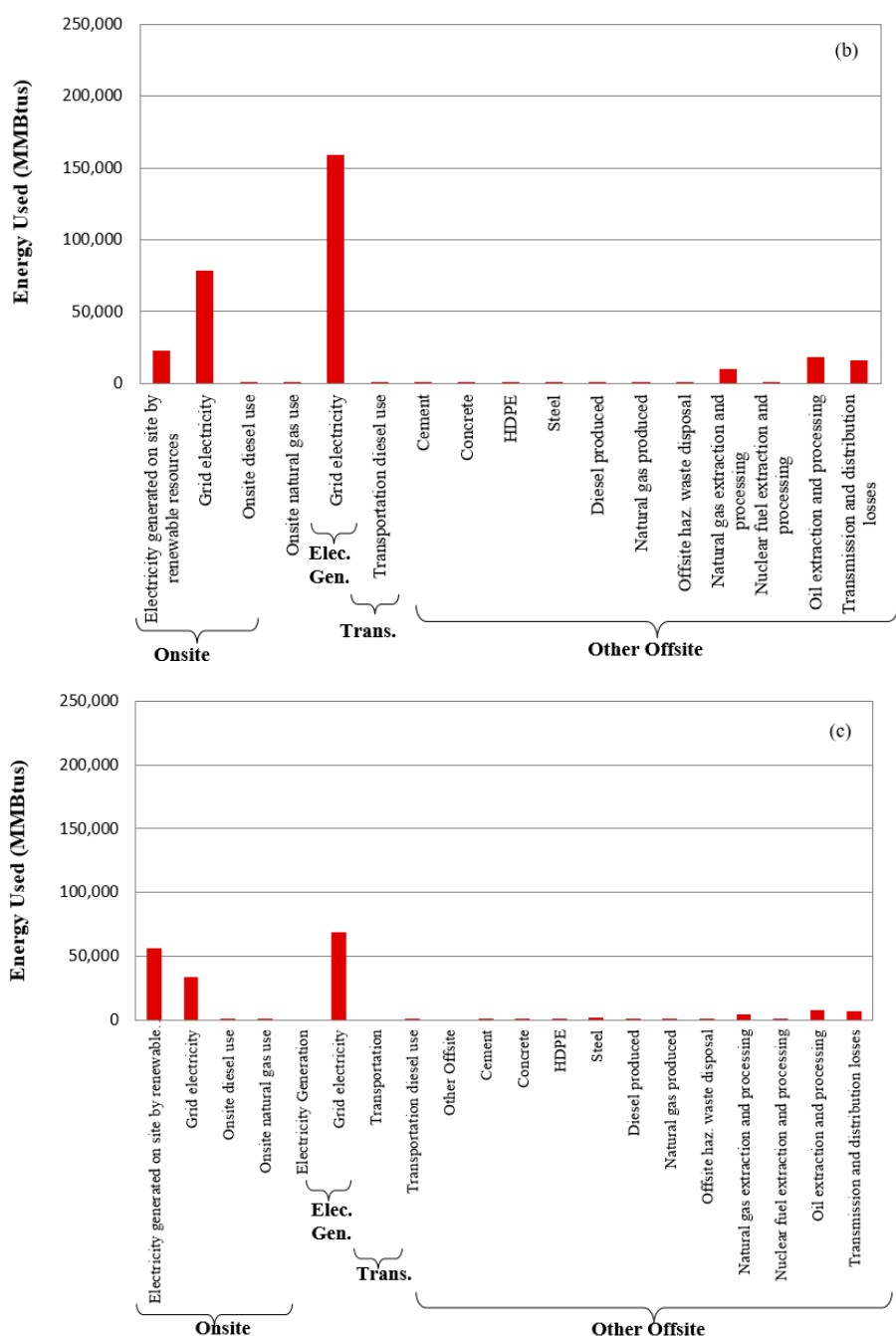

**Figure 5.** Contributors to the total energy used metric; (**a**) Scenario 1, (**b**) Scenario 2 and (**c**) Scenario 3.

From Figures 5–7, it can be observed that the greatest contributor to the environmental footprint is the grid electricity used in remedial operations. One reason of this high contribution is the fuel mix of grid electricity in Pakistan. As discussed earlier, Pakistan mostly relies on conventional fossil fuels such as oil and natural gas for electricity generation. SEFA not only takes into account the amount of fossil fuels used in the electricity generation but also considers the energy which is put into the extraction of these fossil fuels. Thus, the higher the use of fossil fuels, the greater will be their contribution in the overall footprint of the remedy. Integration of onsite renewable energy in cleanup helps in the reduction of electricity usage from grid and consequently lowers the energy demand for fuel extraction and electricity transmission as shown in Figure 5b,c.

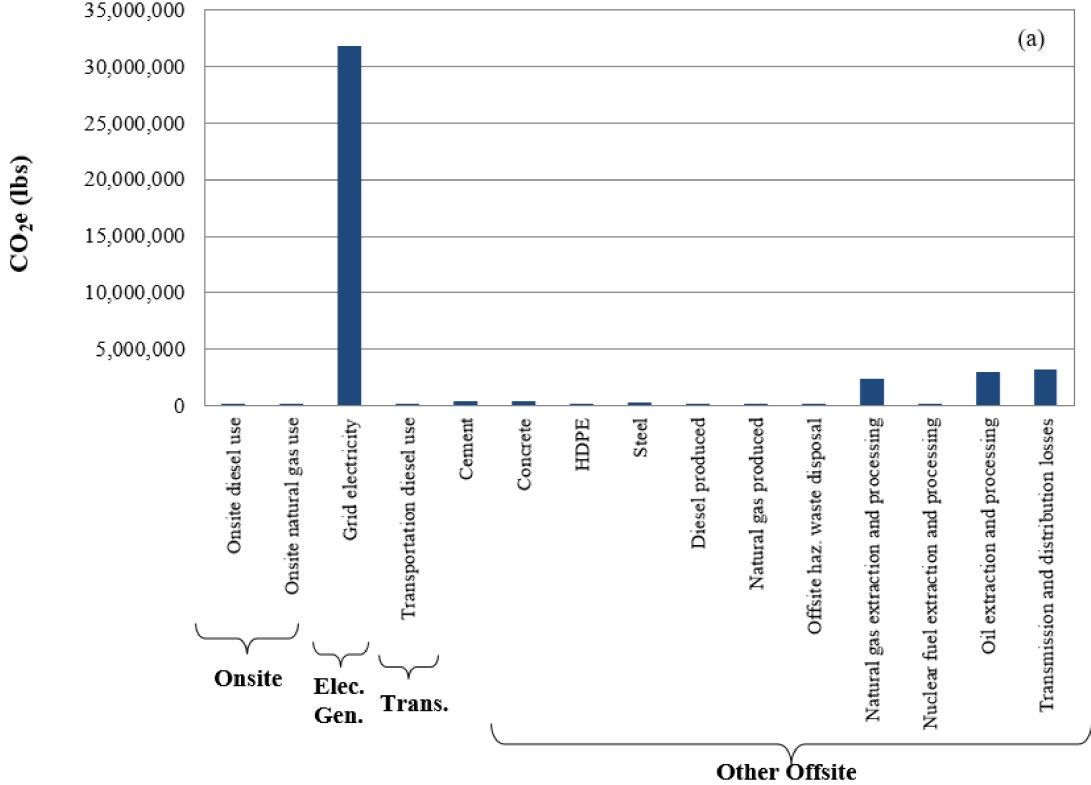

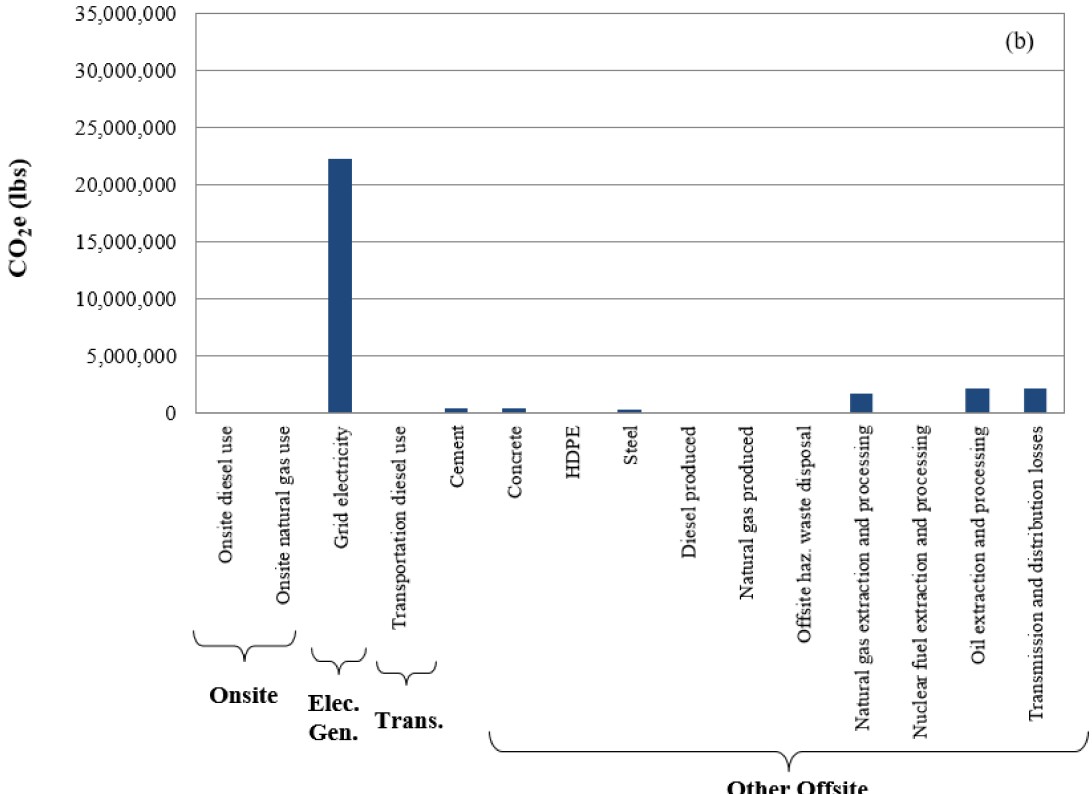

**Figure 6.** *Cont.*

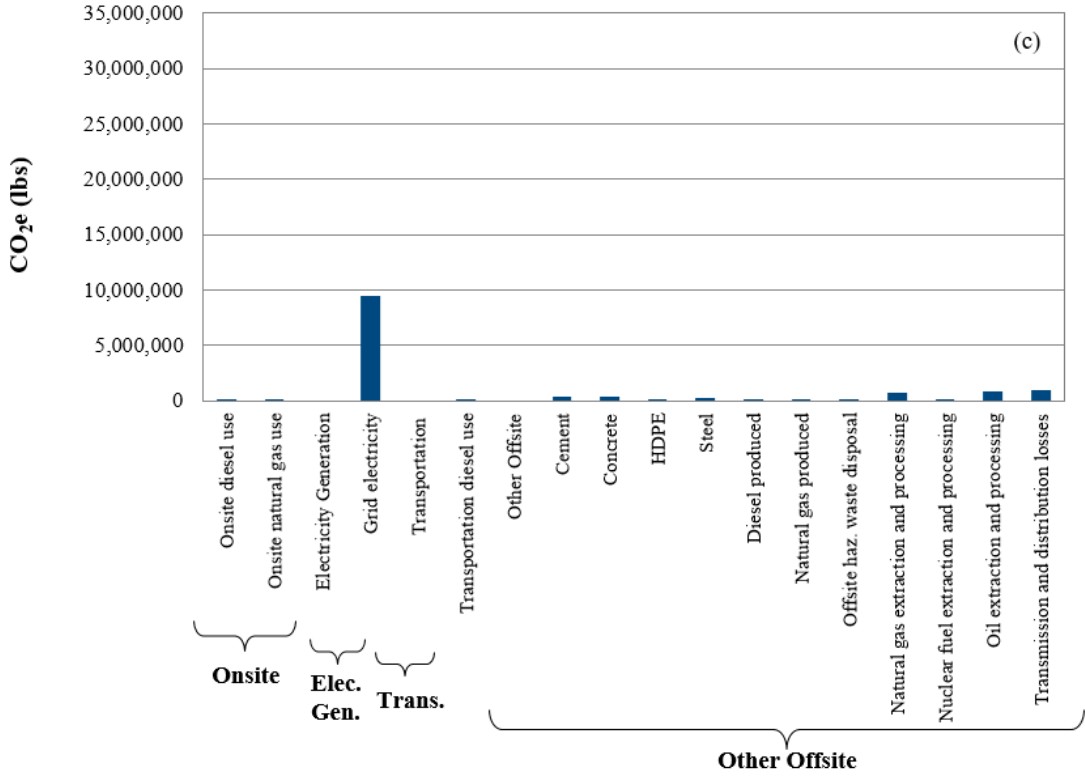

**Figure 6.** Contributors to the CO2 emissions; (**a**) Scenario 1, (**b**) Scenario 2 and (**c**) Scenario 3.

**Table 4.** Reduction in emissions via renewable energy integration in the remedial system.

| Scenario | CO$_2$ Emissions | | Total HAP Emission | | Total NOx, SOx, PM Emissions | |
|---|---|---|---|---|---|---|
| | tons | % Reduction | lbs | % Reduction | lbs | % Reduction |
| 1 | 41,741,757 | - | 1119 | - | 219,726 | - |
| 2 | 29,609,307 | 29 | 801 | 28 | 155,511 | 29 |
| 3 | 13,432,707 | 68 | 377 | 66 | 69,891 | 68 |

The CO$_2$ emissions are also high in the first scenario, as shown in Figure 6a, owing to the fact that only grid electricity is used in the remediation process. The overall behavior of CO$_2$ emissions is nearly the same as of Figure 5, and it is found to have almost negligible emissions in terms of fossil fuel extraction and electricity transmission when renewable energy is incorporated into the remedial system.

The NOx, SOx, HAP and PM emissions are presented in terms of onsite operations, electricity generation, transportation and other offsite operations. It can be observed that the electricity generation and other offsite operations are major contributors in these emissions. The SOx emissions are found to have maximum contribution among all NOx, SOx, PM and HAP. This may be due to the fact that SOx will be emitted when the fuel contains sulphur such as coal and oil. Gasoline is extracted from oil and metals are extracted from ore [23]. For the remedial system, the main sources of SOx emissions are the extraction of fossil fuels for grid electricity, fuels used in transportation, extraction of metals because building steel is used in a very large quantity (292,000 lbs), which makes it a dominant contributor in this category [24–26]. Although NOx emissions are quite less when compared to SOx, and, primarily, they are only due to burning of fuels in motor vehicles and electric utilities, they cannot be ignored in the overall environmental footprint analysis.

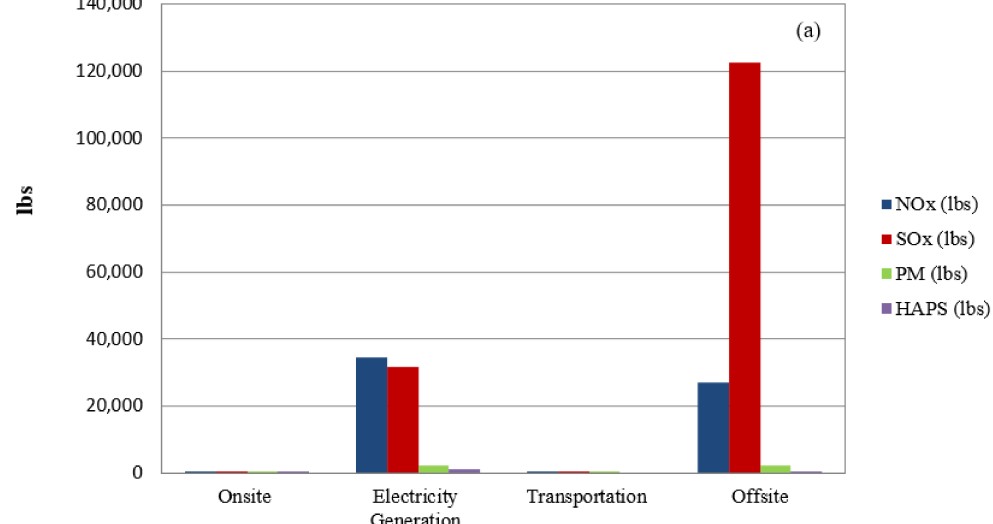

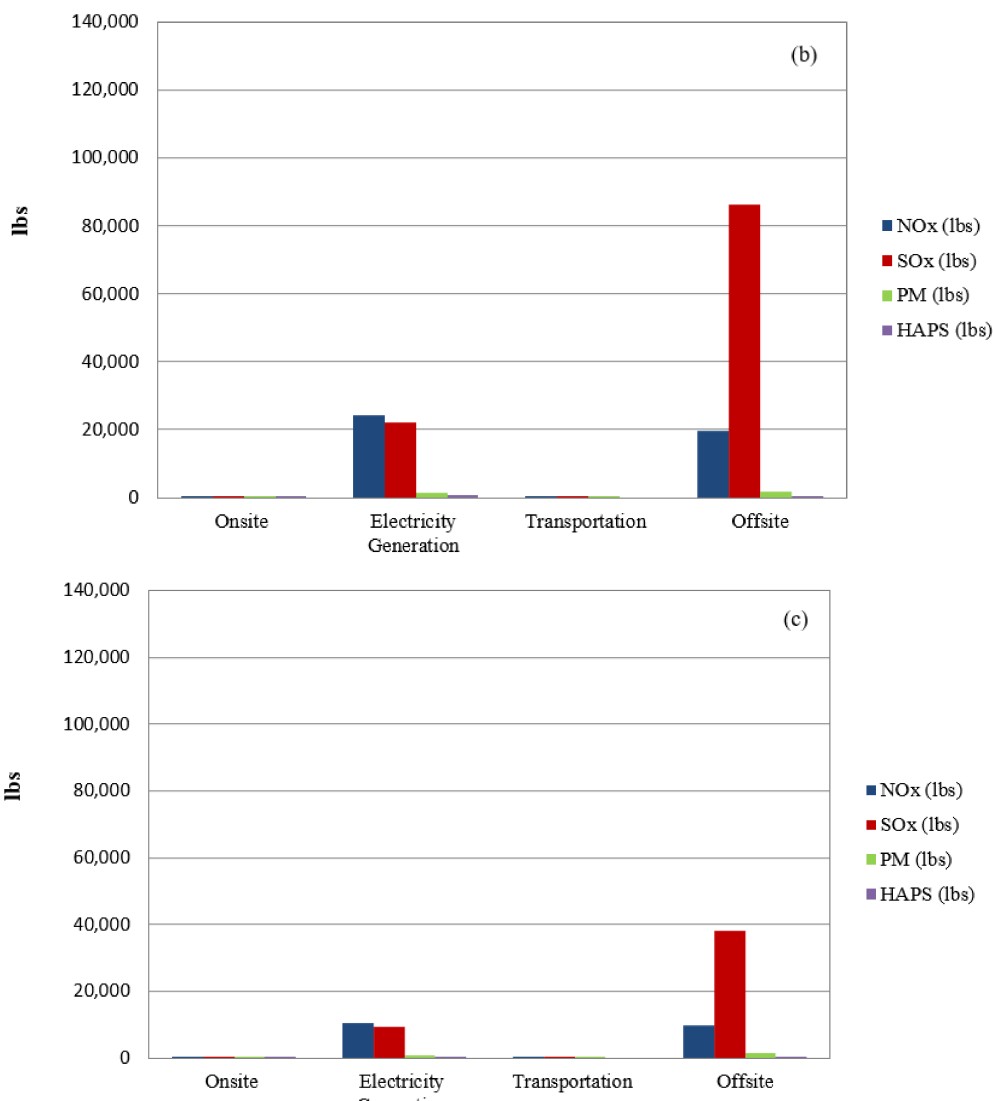

**Figure 7.** Contributors to the total NOx, SOx, PM and HAP emissions; (**a**) Scenario 1, (**b**) Scenario 2 and (**c**) Scenario 3.

Based on the above findings, it is highly recommended to incorporate renewable energy sources not only into the remediation activities but also into the fuel mix of grid electricity in Pakistan. Figures 8 and 9 demonstrate the solar and wind energy potential of Pakistan, respectively, which is quite promising for reducing the overall footprint of any activity in general and, in particular, the remediation system under consideration. For example, the annual average mean daily solar radiation in Karachi is in between 5.1–5.4 kWh/m$^2$ [27], and the wind power class for Karachi is from "Fair" to "Excellent", with a wind speed between 6.2 and 7.8 m/s [28], depending on the region. Thus, installation of renewable energy systems onsite and purchase of electricity from local renewable energy producers for the remediation of contaminated sites in Karachi is a promising idea which can lead to a reduction in the $CO_2$, NOx, SOx, PM and HAP emissions even beyond 68%.

The research methodology used in this study has limitations. The most significant is that it is a single case study on a hypothetical contaminated site and uses a particular data set for analysis. Therefore, it has potential limitations for systematic generalization. Moreover, the results obtained are based on the share of renewable energy resources in each cleanup scenario and could lead to a completely different set of results if the use of renewable energy resources is increased for more sustainability or decreased due to their limited availability.

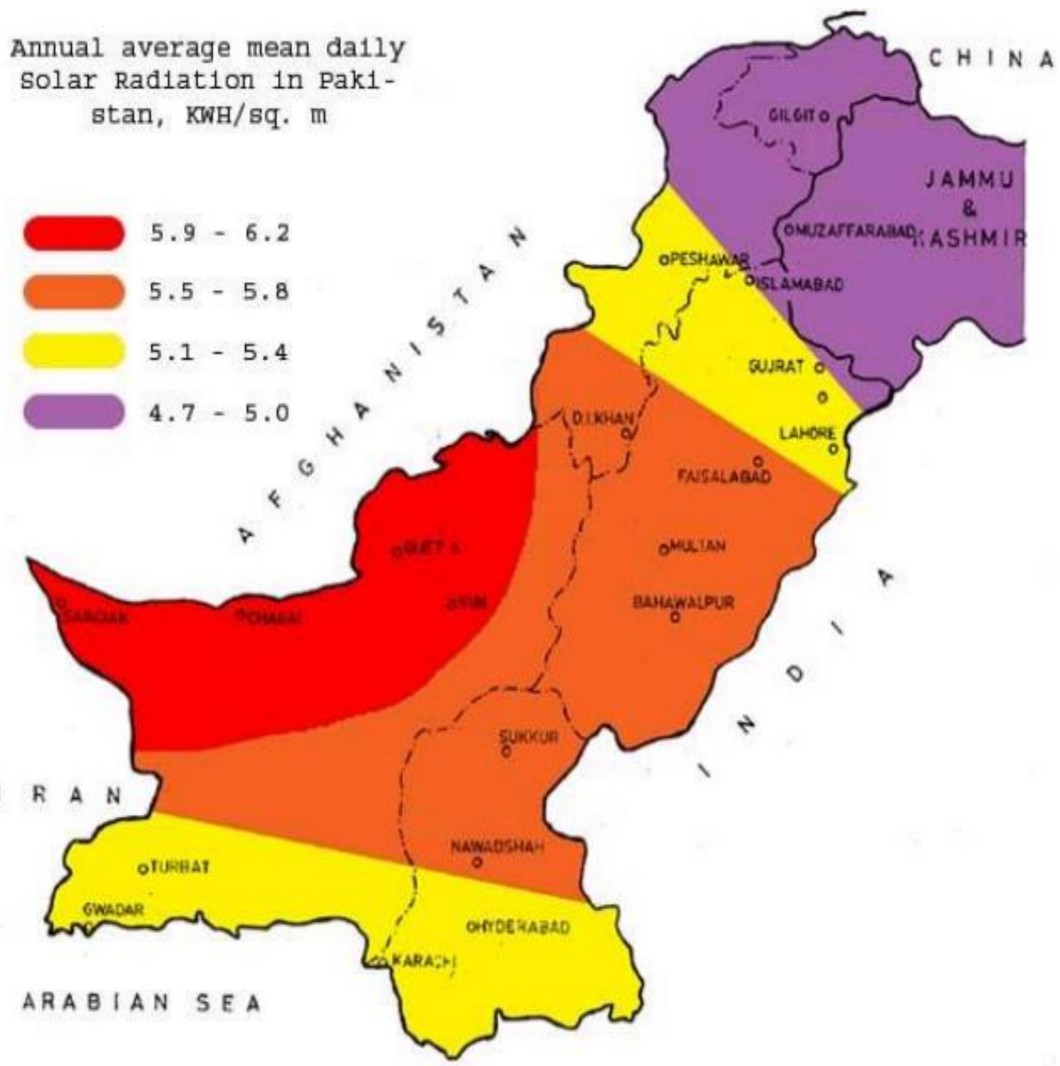

**Figure 8.** Solar energy potential of Pakistan [27].

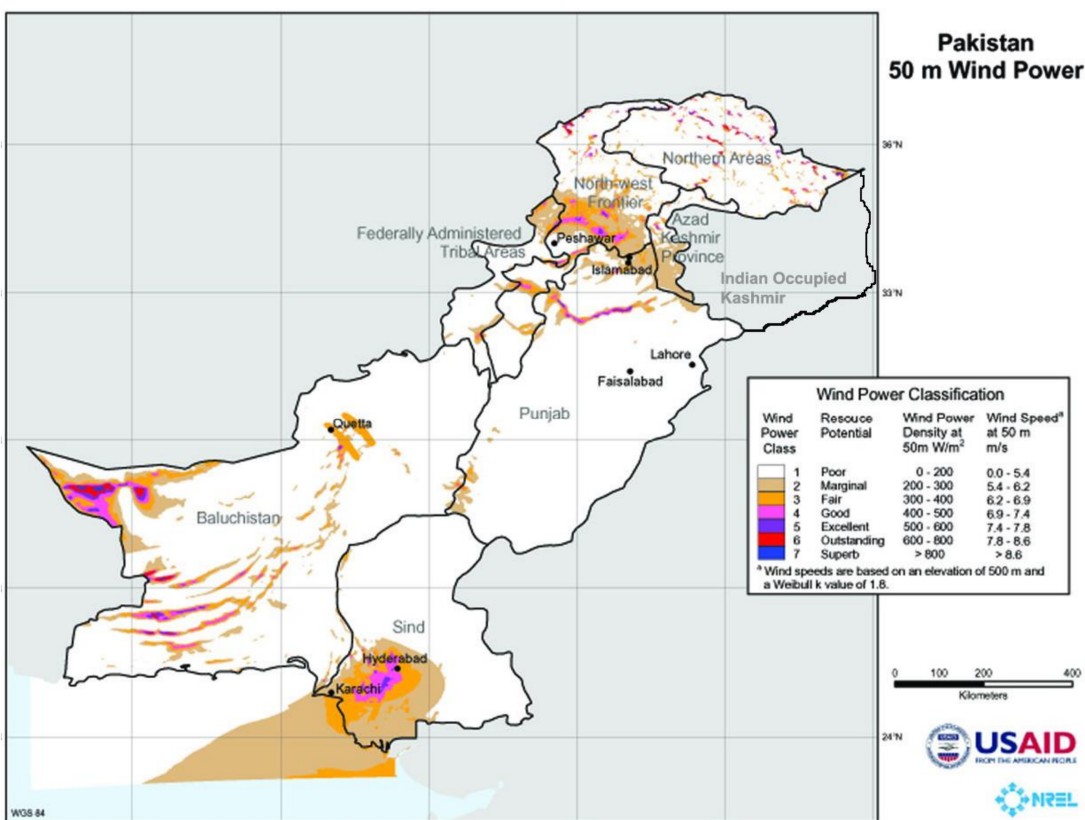

**Figure 9.** Wind energy potential of Pakistan [28].

## 6. Conclusions and Future Work

The environmental footprint of a cleanup at a hypothetical contaminated site is analyzed by using EPA's Spreadsheet for Environmental Footprint Assessment (SEFA). The effect of renewable energy integration into the remedial system is studied by considering three scenarios: cleanup without any renewable energy sources at all, cleanup with a small share of renewable energy sources, and cleanup with a large share of renewable energy sources. It is observed that the greatest contributor in the overall environmental footprint for this cleanup activity is grid electricity due to the fuel mix used in Pakistan. However, integration of renewable energy systems onsite and voluntary purchase of renewable energy can reduce the $CO_2$, $NO_X$, SOx, PM and HAP emissions up to 29% if done on a small scale, and up to 68% if done on a large scale.

The study presented here can be extended further by taking an actual contaminated site and applying this methodology to analyze the environmental footprint of remediation with possible shares of renewable energy resources. Owing to abundant solar resources and excellent wind power class in Pakistan, there is an immense need to incorporate these renewable energy sources in the remediation and to accurately predict the realistic reduction in $CO_2$, $NO_X$, SOx, PM and HAP emissions, thereby promoting green remediation across the country.

**Author Contributions:** Conceptualization, M.A.A.K., Z.Q., and M.A.; methodology M.A.A.K., M.A. and soft-ware, M.A.A.K., Z.Q.; validation, M.A.; formal analysis, M.A., Z.Q. investigation, M.A.A.K., Z.Q. and M.A. resources, A.Z.K.; data curation, M.A.P.M.; writing—original draft preparation, M.A.A.K., Z.Q. and M.A. funding acquisition, M.A.P.M. All authors have read and agreed to the published version of the manuscript.

**Funding:** This research received no external funding.

**Institutional Review Board Statement:** Not applicable.

**Informed Consent Statement:** Not applicable.

**Data Availability Statement:** Not applicable.

**Conflicts of Interest:** The authors declare no conflict of interest.

## Nomenclature

| Symbols | Definition |
|---------|------------|
| kW | Kilo Watt |
| kWh | Kilo Watt Hour |
| HP | Horsepower |
| Btu | British thermal unit |
| gptm | Gallons per ton-mile |
| mpg | Miles per gallon |
| gal | Gallon |
| MMBtu | Metric Million British Themral Unit |
| MWh | Mega Watt Hour |
| lbs | Pound |

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
