# Peer review of "Environmental Footprint Assessment of a Cleanup at Hypothetical Contaminated Site"

_applsci, doi:10.3390/app11114907_

Round 1
Reviewer 1 Report
Dear authors
This topic is not a complete novelty; however, its approach shows an interesting way to analyse the environmental footprint of a contaminated site.
In my opinion, the keywords are not a meaningful and accurate representation of the work developed in the article. I suggest reformulating, in order to give to the reader the nearest information about the topics in the paper.
A complete list of nomenclature is needed.
In the introduction, is necessary to include works related to the use of Spreadsheet for Environmental Footprint Assessment, in order to see the state of the art of this topic.
Regarding the methodology session, the structure is not clear at all, and it’s not easy reading. A restructuring of sessions 2, 3 and 4 are needed.
About the results session, is needed also a restructuring because tables and figures are difficult to follow in the text.
Conclusions need to be in concordance with the aims and allow formulating a set of interesting forwards.
Regarding references, please elaborate on the session, please make a deep revision about the SEFA application
Now, some details:
1) Table 1 is not easy to check.
2) Tables 3 and 4 are not well numbered
3) What is the relevance of Land and Ecosystem topic in the work
Author Response
Reviewer 1 |
Response |
Dear authors |
We thank the reviewer for this encouraging comment. Although the idea is not completely novel, very limited published literature is available about using SEFA for environmental impact assessment of remediation of contaminated sites. Hence, this paper is an effort to develop a methodology for analyzing the environmental impact of a cleanup of any contaminated site using this easily accessible tool. |
In my opinion, the keywords are not a meaningful and accurate representation of the work developed in the article. I suggest reformulating, in order to give to the reader the nearest information about the topics in the paper. |
The new keywords are as follows. Environmental footprint; cleanup; green remediation; renewable energy sources |
A complete list of nomenclature is needed. |
Thank you for the constructive comment. We added the list of Nomenclature before references section. |
In the introduction, is necessary to include works related to the use of Spreadsheet for Environmental Footprint Assessment, in order to see the state of the art of this topic. |
Yes, we have added the relevant studies in the introduction section. |
Regarding the methodology session, the structure is not clear at all, and it’s not easy reading. A restructuring of sessions 2, 3 and 4 are needed. |
Thank you very much for highlighting this point. The methodology and sections 2,3 and 4 are structured accordingly and in a sequence. |
About the results session, is needed also a restructuring because tables and figures are difficult to follow in the text. |
Thank you very much for highlighting this point. The result section is structured accordingly. |
Conclusions need to be in concordance with the aims and allow formulating a set of interesting forwards. |
The section on Conclusions and Future work is revised carefully as per the reviewer’s comments. |
Regarding references, please elaborate on the session, please make a deep revision about the SEFA application |
As mentioned earlier, very limited literature is available related o application of SEFA. However, the authors are fortunate to find a very good paper which is summarized in the introduction section of the paper as follows. “Marco et al. [29] studies the environmental impact of the remediation of an aquifer below an industrial site in the Bologna area. Three proposed systems were investigated for environmental impact using two environmental footprint analysis tools i.e. SiteWiseTM and SEFA. The three solutions studies and compared for environmental footprint are (i) groundwater extraction system, treatment and reinjection, (ii) reductive bioremediation and (iii) in situ chemical oxidation (ISCO). Based on the results obtained from both tools, bioremediation is found to be the appropriate remediation with minimum GHG and lower levels of environmental impacts. Also, the higher environmental impacts caused by ISCO is due to frequent multiple injection events with Potassium Permanganate in contrast with the general single injection performed with bioremediation.” |
Now, some details: |
|
1) Table 1 is not easy to check. |
1) Table 1 all the sections are splitted row wise for better clarity, we have also improved the figure quality. 2) Tables 3 and 4 are numbered properly in the revised manuscript. 3) This metric is about a qualitative description of potential disturbance in land and eco-system which will be caused by the employed remediation technique. Although, no such disturbance is caused in the remedial action proposed in this work, it is included in Table 2 and Table 3 for completeness. |
|
|

Reviewer 2 Report
The manuscript entitled “Environmental Footprint Assessment of a Cleanup at Hypothetical Contaminated Site” presents a research related to the management and remediation techniques of contaminated areas. The authors presented a methodology to quantify the environmental footprint of a cleanup for a hypothetical contaminated site by using spreadsheet prepared by the Environmental Protection Agency. The assessment was made regarding three different scenarios covering cleanup with no renewable energy sources, cleanup with a small share of renewable energy sources, and cleanup with a large share of renewable energy sources.
General comments:
• Explain the abbreviation EPA in the abstract. I know that it is commonly known but all abbreviations used must be explained in the place where they have been used for the first time.
• Lines 86-90 are redundant. It is not necessary to summarize the content of the subsequent chapters. The same concerns lines 178-180.
• The quality of Figure 2 should be improved. Please add a scalebar an the north direction.
• The authors stated that “two provinces Khyber Pakhtunkhwa (KPK) and Punjab are found to have maximum concentration of contaminated sites as shown in the map of Pakistan”. I cannot see these provinces on the map. Please mark them clearly.
• The authors mentioned the LNAPL caused by a spill from an underground tank. Thus, regarding LNAPL features, I expected to see that LNAPL floats on the water table and does not penetrate below the water table. The presentation of the plume in Figure 3 is a bit confusing. Please also show the tank in the Figure to express the source of contamination.
• Please use SI units, for example meter instead of feet, kg instead of lbs. Please unify the system of unit presentation in accordance with the SI system.
• Line 324: Summary of the Environmental Footprint of cleanup under all scenarios” should be Table 3.
• Line 326: “Reduction in emissions via renewable energy integration in remedial system” should be Table 4.
• Figure 5: there are (a) and (c) presented. Where is (b)? If the distinction of (a), (b), (c) is required, it is also necessary to explain this in the caption of the Figure.
• Figure 6: “Other offsite” is not visible in (a). Where is (b). What do the (a), (b), (c) refer to?
• Figure 7: (c) is missing.
• Section 6. Conclusion and Future works should be thoroughly edited and corrected. This chapter should summarize the most important findings of performed research. The Figures 8 and 9 do not fit to this section. They should be rather incorporated to the Results and Discussion. The discussion of the results in connection to the literature references cited in section 6 should be transferred to Results and Discussion.
• Please specify what is the novelty in presented study.
• I expect to see the limitations of performed study in the Discussion.
• The theme of the manuscript is valuable, nevertheless the content must be better arranged to attract the readers’ attention.
Author Response
Reviewer 2 |
Response |
• Explain the abbreviation EPA in the abstract. I know that it is commonly known but all abbreviations used must be explained in the place where they have been used for the first time. |
The abbreviation EPA is explained in the abstract of revised manuscript. |
• Lines 86-90 are redundant. It is not necessary to summarize the content of the subsequent chapters. The same concerns lines 178-180. |
Lines 86-90 have been removed in the revised manuscript. Lines 178-180 have been removed in the revised manuscript. |
• The quality of Figure 2 should be improved. Please add a scale bar and the north direction. |
The quality of Figure 2 has been improved. A scale bar and north direction are also added to the Figure. |
• The authors stated that “two provinces Khyber Pakhtunkhwa (KPK) and Punjab are found to have maximum concentration of contaminated sites as shown in the map of Pakistan”. I cannot see these provinces on the map. Please mark them clearly. |
All provinces i.e. KPK, Punjab, Sindh and Baluchistan are now indicated in the map shown in Figure 2. |
• The authors mentioned the LNAPL caused by a spill from an underground tank. Thus, regarding LNAPL features, I expected to see that LNAPL floats on the water table and does not penetrate below the water table. The presentation of the plume in Figure 3 is a bit confusing. Please also show the tank in the Figure to express the source of contamination. |
We completely understand reviewer concern that LNAPL should float on the water table. This is true and figure 3 is modified to avoid any confusion. In the revised version, Figure 3 (a) represents the LNAPL release and subsequent migration, whereas, Figure 3 (b) shows the design of remedial system. It is easy to understand that the plume is showing the dissolved LNAPL which is included in the figure just to represent the depth from which the remedial system is working on the cleanup process. |
• Please use SI units, for example meter instead of feet, kg instead of lbs. Please unify the system of unit presentation in accordance with the SI system. |
The spreadsheet available for analysis has units in FPS system. It is very difficult to change all units to SI system, and at the same time, changing the units also introduce a source of error in the values or interpretation of results. We request reviewer to understand this situation and allow to use FPS system of units in the manuscript. |
• Line 324: Summary of the Environmental Footprint of cleanup under all scenarios” should be Table 3. |
Table number has been corrected in the revised manuscript. |
• Line 326: “Reduction in emissions via renewable energy integration in remedial system” should be Table 4. |
Table number has been corrected in the revised manuscript. |
• Figure 5: there are (a) and (c) presented. Where is (b)? If the distinction of (a), (b), (c) is required, it is also necessary to explain this in the caption of the Figure. |
Figure 5 (a), (b) and (c) are labelled properly in revised manuscript. The explanation of (a), (b) and (c) is also provided in the caption of the figure. |
• Figure 6: “Other offsite” is not visible in (a). Where is (b). What do the (a), (b), (c) refer to? |
Figure 6: “other offsite” is visible in (a) in the revised manuscript. 6 (b) is labelled properly. The explanation of (a), (b) and (c) is also provided in the caption of the figure. |
• Figure 7: (c) is missing. |
Figure 7 (c) is now included in the revised manuscript. |
• Section 6. Conclusion and Future works should be thoroughly edited and corrected. This chapter should summarize the most important findings of performed research. The Figures 8 and 9 do not fit to this section. They should be rather incorporated to the Results and Discussion. The discussion of the results in connection to the literature references cited in section 6 should be transferred to Results and Discussion. |
The conclusion and future work section is revised as per the reviewer’s comments. Figures 8 and 9 are moved to results and discussion as advised by the reviewer. Following addition has been made to section on conclusion and future work. “The study presented here can be extended further by taking an actual contaminated site and applying this methodology to analyze the environmental footprint of remediation with possible shares of renewable energy resources. Owing to abundant solar resources and excellent wind power class in Pakistan, there is an immense need to incorporate these renewable energy sources in the remediation and to accurately predict the realistic reduction in CO2, NOX, SOx, PM and HAP emissions, thereby, promoting green remediation across the country.” |
• Please specify what the novelty is in presented study. |
Although the idea is not completely novel, very limited published literature is available about using SEFA for environmental impact assessment of remediation of contaminated sites. Hence, this paper is an effort to develop a methodology for analyzing the environmental impact of a cleanup of any contaminated site using this easily accessible tool. |
• I expect to see the limitations of performed study in the Discussion. |
The limitations are added in the discussion section as follows. “The research methodology used in this study has limitations. The most significant is that it is a single case study on a hypothetical contaminated site and uses a particular data set for analysis. Therefore, it has potential limitations for systematic generalization. Moreover, the results obtained are based on the share of renewable energy resources in each cleanup scenario and could lead to a completely different set of results if the use of renewable energy resources is increased for more sustainability or decreased due to their limited availability.” |
• The theme of the manuscript is valuable, nevertheless the content must be better arranged to attract the readers’ attention. |
We thank the reviewer for this encouraging comment. We tried our best to arrange the content in a more meaningful manner in the revised manuscript. |
|
|

Round 2
Reviewer 2 Report
Dear authors, I appreciate your work and time devoted to improve the manuscript. Almost all my concerns have been addressed, but I still encourage you to convert units to SI system. As the Applied Sciences is an international journal directed to readers from all over the world, please facilitate interpretation your results by using units that are commonly recommended and preferred. It seems that in terms of units, the manuscript is a bit chaotic. For example you use feet and meters in the same work, it is confusing.
The quality of Figure 8 should be also improved, the names of cities presented on the Figure are not readable now.
Author Response
Respected Reviewer, I hope you are doing well and safe. We would like to request you regarding the additional comment concerned about changing units. We would like to notify you that changing units is not that easy and it can introduce a lot of errors in the results reported, therefore, kindly consider accepting the results as they are presented in the manuscript. Furthermore, we are ready to work on any suggestion, but, this is something that is too confusing for the authors and the results are generated by SEFA EPA. Any changes in the excel files can lead to major errors in the results. Kind Regards, Zakria Qadir Corresponding Author